# Proteins Found in the Triple-Negative Breast Cancer Secretome and Their Therapeutic Potential

**DOI:** 10.3390/ijms24032100

**Published:** 2023-01-20

**Authors:** Peter R. McHenry, Jenifer R. Prosperi

**Affiliations:** 1Department of Natural Science and Health, Faculty of Arts and Sciences, Middle East University, P.O. Box 90-481, Jdeidet El Metn 1202-2040, Lebanon; 2Department of Biochemistry and Molecular Biology, Indiana University School of Medicine-South Bend, South Bend, IN 46617, USA; 3Department of Biological Sciences, University of Notre Dame, Notre Dame, IN 46556, USA

**Keywords:** TNBC, breast cancer, secretome, angiogenesis, cytokine, ECM, protease, tumor microenvironment, therapeutic target

## Abstract

The cancer secretome comprises factors secreted by tumors, including cytokines, growth factors, proteins from the extracellular matrix (ECM), proteases and protease inhibitors, membrane and extracellular vesicle proteins, peptide hormones, and metabolic proteins. Secreted proteins provide an avenue for communication with other tumor cells and stromal cells, and these in turn promote tumor growth and progression. Breast cancer is the most commonly diagnosed cancer in women in the US and worldwide. Triple-negative breast cancer (TNBC) is characterized by its aggressiveness and its lack of expression of the estrogen receptor (ER), progesterone receptor (PR), and HER2, making it unable to be treated with therapies targeting these protein markers, and leaving patients to rely on standard chemotherapy. In order to develop more effective therapies against TNBC, researchers are searching for targetable molecules specific to TNBC. Proteins in the TNBC secretome are involved in wide-ranging cancer-promoting processes, including tumor growth, angiogenesis, inflammation, the EMT, drug resistance, invasion, and development of the premetastatic niche. In this review, we catalog the currently known proteins in the secretome of TNBC tumors and correlate these secreted molecules with potential therapeutic opportunities to facilitate translational research.

## 1. Introduction

Secreted factors released from primary tumors are able to alter the tumor microenvironment and, through both autocrine and paracrine mechanisms, the secretome of the tumor itself [1]. Collectively, these secreted factors (including proteins, RNAs, extracellular vesicles, etc.) make up the “secretome”. The tumor cell secretome generally comprises cytokines, growth factors, proteins from the extracellular matrix (ECM), proteases and protease inhibitors, membrane and extracellular vesicle proteins, peptide hormones, and metabolic proteins (Figure 1). This variety of secreted proteins renders the tumor cell secretome as an obvious mechanism by which tumor cells can promote chemoresistance, induce metastasis, and regulate the immunological response.

Breast cancer is the leading cause of cancer in women in the US and worldwide. Breast cancer can be subdivided into molecular subtypes that vary by their gene expression, prognosis, and treatment options. Triple-negative breast cancer (TNBC) is clinically challenging because, despite its initial favorable response to standard chemotherapeutic regimens, it often becomes resistant. This subtype of breast cancer, named for its lack of expression of the estrogen receptor (ER), progesterone receptor (PR), and HER2, is not amenable to targeted therapy directed at the ER and HER2, leaving patients to rely on chemotherapy as the standard of care [2,3]. TNBC also has the lowest 5-year survival rate among all the subtypes, demonstrating a need for new therapeutic options for these patients. Although patients initially respond well to chemotherapy, 50–80% of women with TNBC will relapse or develop resistance to the chemotherapeutic agent, making this a major driver of breast cancer mortality. Therefore, there is an immediate unmet need to identify and characterize other molecular events and downstream pathways important for the initiation of TNBC, resistance to chemotherapy, and recurrence. One aspect that may influence the development of chemoresistance in TNBC is the interaction of secreted factors with the tumor cells themselves or with the surrounding tumor microenvironment (TME).

Previous reviews of the TNBC secretome have focused on the mechanisms of chemoresistance [4], extracellular vesicles [5], the induction of a pro-inflammatory microenvironment [6], induction of metastatic spread [7], or a newly discovered autocrine loop involving HMGA1 ([8] and reviewed in [9]). Secreted molecules include proteins and a variety of RNA species (noncoding regulatory RNAs as well as protein-coding circular mRNAs). However, secreted RNAs in the context of TNBC have been recently reviewed elsewhere [5,10]. In this review, we intend to catalog the currently known proteins in the secretome of TNBC tumors and correlate these secreted molecules with potential therapeutic opportunities to facilitate translational research.

## 2. The TNBC Secretome

### 2.1. Cytokines and Growth Factors

Ziegler et al. (2016) used mass spectrometry to identify proteins secreted from three TNBC cell lines (DT22, DT28, and MDA-MB-231) compared to a luminal-type cell line with normal expression of the ER, PR, and HER2/Neu (MCF-7) and a non-tumor cell line. Spectral counts were used as a proxy of protein expression level in conditioned medium (CM). Using this measure, the authors found that the immune-modulating proteins SAA1, thrombospondin (THBS1), and growth factor TGFβ1 were more highly expressed in TNBC CM on average compared to MCF-7 CM [11]. This study identified many proteins enriched in the TNBC secretome, and we will refer to the study again throughout this review.

Among the cytokines and growth factors secreted by TNBC (Table 1), a theme of blood vessel regulation stands out. For example, BDNF is a growth factor that promotes tumor and endothelial cell migration [12,13]. SLIT3 regulates axon guidance in the brain but also organizes angiogenic processes [14]. On the other hand, SLIT3 produced by CD36^+^ stromal fibroblasts was one of the primary secreted factors that suppressed the growth of TNBC cells in vitro [15]. Other factors of the TNBC secretome, including PLGF, VEGF, TXNIP, and CXCL1/GRO, also regulate angiogenesis [7,12]. END1 secreted by TNBC regulates vasoconstriction [16]. Further, the expression of proteins involved in angiogenesis (including FGF proteins and TGFβ) increased in TNBC cells when the anti-metastatic protein EPAC1 was knocked down using siRNA [17].

Vascular endothelial growth factor (VEGF) itself was detected in the TNBC secretome in at least three studies. The source of VEGF varied, including TNBC cells depleted of Syndecan-1 [16], interstitial fluid of TNBC xenograft tumors in mice [28], and adipose-derived mesenchymal stem cells treated with CM from TNBC cells [6]. The ubiquity of this angiogenesis regulator in the TNBC secretome, as well as the presence of a constellation of secreted factors that regulate its activity (discussed above), seems to indicate that it is crucial in TNBC disease progression.

Multiple studies found that TGFβ was secreted by TNBC cells. However, the role of TGFβ in angiogenesis remains unclear (e.g., it reduces VEGF-A signaling [29] but actually works with VEGF to modulate angiogenesis in a manner that is dependent on the TGFβ receptor subclass [30]. Another member of the TGFβ superfamily, NODAL, was found in the TNBC secretome [27]. NODAL is a fetal development protein that is reactivated in multiple cancer types, where it stimulates the angiogenic activity of VEGF [31,32]. In addition, NODAL was found to activate fibroblasts associated with cancer cells and further alter the tumor microenvironment [27,33].

Another theme of the TNBC secretome is regulation of the immune system. A wide assortment of immune-modulating proteins were detected in the TNBC secretome. For example, the secretome of TNBC cells (MDA-MB-231) induced the expression of the pro-inflammatory cytokines CCL2, CCL5, IL-1β, and IL-6 in adipose-derived mesenchymal stem/stromal cells. The authors went on to show that the IL-6 secreted by the TNBC cells was responsible for inducing chemotaxis [6]. In agreement with those findings, another group showed that a combination of drugs that suppress both IL-6 and CCL5 signaling reduced TNBC tumor growth and metastasis in a mouse xenograft model [34].

Logue et al. (2018) also showed an increase in cytokine release from MDA-MB-231 cells, in particular IL-6, IL-8, CXCL1/GRO, GM-CSF, and TGFβ2, which were under the control of IRE1. Of the cytokines in that study, CXCL1/GRO was the common factor secreted by four different TNBC cell lines [19]. Suarez et al. (2022) found that the secretome of MDA-MB-231 cells induced the expression of immune modulators COX2, HIF-1α, VEGF-A, and PD-L1 in adipose-derived mesenchymal stem/stromal cells [6].

An interesting protein found in the TNBC secretome is HMGA1. HMGA1 is a chromatin structural protein that typically plays a role in regulating gene transcription. However, recently it was found to bind to the RAGE receptor, earning it a new role as an immune modulator, joining the closely related protein HMGB1 [20,21].

Sayyad et al. (2019) performed a cytokine/chemokine array and showed that GRO-α/CXCL1, ICAM-1, IL-6, IL-8, GM-CSF, and CCL5 were all differentially expressed in MDA-MB-231 cells compared to syndecan-1 knockdown cells. The same study showed that syndecan-1 contributes to TNBC cells’ ability to cross the blood–brain barrier and to the brain metastasis of TNBC cells via cardiac injection in mice [35]. The authors further attempted to disrupt the blood–brain barrier using only IL-6 and IL-8 at relevant concentrations in the culture medium, but these cytokines alone were insufficient, suggesting that a complex milieu of cytokines and other molecules may be needed for brain metastasis.

### 2.2. Extracellular Matrix Proteins

Extracellular matrix (ECM) proteins (Table 2) are an important contributor to the tumor microenvironment. We found that while studies of TNBC-related ECM proteins identified many proteins, different proteins were identified in different studies.

The Ziegler mass spec study found that a large number of ECM proteins (BGN, CD44, CD109, DAG1, DCN, ECM1, EFEMP1, FMOD, IGFBP4, IGFBP7, LTBP1, L1CAM, LGALS1, LGALS3BP, LOXL2, LTBP1, NRCAM, P4HB, PLOD1, PPIB, TGFBI, THBS1, TLN1, and TNC) were more highly expressed in TNBC CM on average compared to in MCF-7 CM, although DCN and TGFBI were also more highly expressed in non-tumor MCF-10A CM [11]. (See above for discussion of TGFβ as a growth factor.)

Jang (2020) found that CD44 was the most differentially released protein in TNBC cells. In addition, CD44 appeared to regulate the release of cytokines from the surrounding macrophage population [22]. CD44 is a multifunctional surface glycoprotein involved in cell adhesion and signaling. CD44 is frequently shed into the extracellular milieu, and serum levels of CD44 have been proposed as a prognostic marker in breast cancer [38]. Another surface glycoprotein, carcinoembryonic antigen (CEA), was discovered to have a complementary role with CD44 in cancer cell adhesion and metastatic potential [39]. Very recently, promising nanoparticle drug delivery systems have been developed that target CD44 and CEA in colorectal cancer cells and TNBC [40,41,42,43,44].

Another surface antigen, CD109, is a GPI-anchored coreceptor and negative regulator of TGFβ [45]. The soluble form of CD109 also binds to TGFβ and inhibits TGFβ signaling [46]. CD109 protein expression is much higher in TNBC than in non-TNBC, and CD109 expression correlates with a higher histological grade and poorer post-operative survival in breast cancer patients [47]. Soluble CD109 shed from breast cancer cells also promoted malignant growth in 3D organotypic culture [48].

Another TNBC secretome protein, ECM1 [11,26], was found to induce angiogenesis [49] and to promote tumor cell proliferation through EGFR signaling, which conferred resistance to trastuzumab [50]. In breast tumors, high ECM1 expression is associated with poor prognosis [51], and ECM1 confers endocrine resistance on ER+ tumors that may be mediated by SRC [52]. ECM1 was also discovered in the secretome of HER2-overexpressing cell lines where it was found to promote endothelial network development in 3D culture [53]. However, little is known about the specific effects of ECM1 on TNBC.

The glycoprotein fibulin 3 (EFEMP1/FBLN3) is an ECM1-interacting protein that was discovered in the TNBC secretome [11]. Fibulin 3 is overexpressed in breast cancers with low HER2 expression, including TNBC tumors [54]. Fibulin 3 has also been found to promote tumor cell invasiveness in TNBC xenografts [55]. In contrast, *inhibition* of fibulin 3 by miR-9 in normal fibroblasts promoted their activation and pro-metastatic effects on TNBC cells [56]. Thus, the role of fibulin 3 in the TNBC secretome is still unclear.

The closely related protein fibulin 1 (FBLN1) is a fibronectin (FN)-interacting protein that was also found in the TNBC secretome [15]. Much of what we know about fibulin 1 with respect to breast cancer involves estrogen signaling. Estrogens upregulate expression of fibulin 1—especially that of the splice variant fibulin 1C [57,58]. Fibulin 1 was inversely associated with cathepsin D expression in an immunohistochemical analysis of breast tumors [59]. Interestingly, both fibulin 1 and cathepsin D were found in the TNBC secretome [60]. Importantly, fibulin 1 may also confer resistance to doxorubicin treatment on breast cancer cells [61].

Among the proteins discovered in the TNBC secretome are several insulin-like growth factor-binding proteins (IGFBPs). Members of this family of proteins bind to and increase the half-life of IGF [62]. However, these proteins have diverse functions. One family member, Cyr61 (also called CCN1 and IGFBP10), is expressed in TNBC cells and tumors, where it interacts with the urokinase plasminogen activator receptor (uPAR) [63]. Cyr61 has long been known to promote angiogenesis and tumor growth [64]. High levels of Cyr61 correlated with relapse in TNBC patients, and silencing of Cyr61 reduced the invasiveness of TNBC cell lines and reduced tumor burden and microvascular density in xenograft mouse TNBC tumors [65]. Another family member identified in the TNBC secretome, IGFBP7, was implicated in angiogenesis in ovarian cancer independent of VEGF [66].

In addition to these specific ECM proteins, Lee (2014) found α2,3-sialylated N-glycoproteins (ECM proteins that bind lectins) expressed in the conditioned medium of TNBC cell lines [67]. In a study on sialic acid glycosylation in breast cancer, expression of polysialic acid (preferentially expressed during fetal development) positively correlated with an invasive phenotype of breast cancer cell lines and with TNM staging of patient tumors. And in the same study, knockdown of sialyl transferase X (STX) reduced the motility of MDA-MB-231 cells [68]. These studies serve to remind us that the identity of proteins in the ECM is only part of the story, since many ECM proteins undergo extensive post-translational modifications, which may also be targetable.

### 2.3. Proteases and Protease Inhibitors

The TNBC secretome contains many enzymes that activate, modify, or destroy other proteins in the extracellular space. Here, we discuss a number of these proteases along with their inhibitors (Table 3).

In the Ziegler study mentioned above, proteases CTSZ, GGH, and PCSK9 were more highly expressed in TNBC CM on average compared to in MCF-7 CM. They also found that the protease inhibitors APLP2, APP, PI3, SERPINE1, TIMP1, and TIMP2 were more highly expressed in TNBC CM on average compared to in MCF-7 CM [11].

Cathepsin D (CTSD) is an aspartic protease typically found in the lysosome. However, the unprocessed proenzyme of CTSD can be “derouted” into endosomes and subsequently secreted. In the acidic tumor microenvironment, CTSD undergoes autoactivation. High extracellular expression of CTSD is associated with metastasis in breast cancer [69]. The pro-form of CTSD also binds to various proteins and has other cell signaling roles [70]. CTSD can stimulate cancer growth and progression not only by its catalytic activity on inflammatory cytokines and ECM proteins, but also by its non-catalytic activity linked to angiogenesis [71]. Therapeutic targeting of CTSD has been an area of intense study. Anantaraju (2016) identified several small-molecule inhibitors that blocked CTSD activity and inhibited breast cancer cell growth [72]. Other inhibitors have been described as well [73,74,75]. In addition, monoclonal antibody therapy against CTSD in TNBC xenograft tumors showed promising results; the antibodies targeted the tumors, elicited an immune response, and showed low toxicity [76].

Cathepsin Z (CTSZ) is a cysteine protease that has been little-studied in breast cancer, but its overexpression in hepatocellular carcinoma increased the EMT and the expression of proteins involved in matrix remodeling. Expression of CTSZ also correlated with an advanced clinical stage [77].

Several of the proteases found in the TNBC secretome are blood-clotting factors. Thromboembolism associated with cancer (Trousseau syndrome) has been recognized since the 19^th^ century and is a leading cause of cancer mortality. It is most common in lung and visceral tumors but can also occur in breast cancer patients [78,79,80]. The coagulation factors plasminogen (PLG) [26], tissue plasminogen activator (PLAT) [26], urokinase plasminogen activator (uPA/PLAU) [25] [28]), and tissue factor (TF) [16] (transmembrane glycoprotein) are all secreted from TNBC cells or xenograft tumors. This is unsurprising since the plasminogen–plasmin system has long been known to regulate angiogenesis in both normal and disease states [81]. A soluble form of the urokinase receptor (suPAR/PLAUR) is also secreted by TNBC cells and appears to act as a cytokine with a variety of functions [25]. In addition to its blood-clotting activity, TF has been associated with malignancy and angiogenesis in various cancer types [82]. Indeed, several strategies to inhibit TF activity in TNBC have already been investigated [83,84,85]. As described above, we found that the plasminogen–plasmin system was implicated in at least four independent studies of the TNBC secretome. Taken together, these data suggest a role for blood coagulation factors in the progression and pathology of TNBC disease.

Amyloid-β precursor protein (APP) and amyloid-β precursor-like protein 2 (APLP2) are related protease inhibitor proteins most closely associated with Alzheimer’s disease. The secreted form of APP is also called protease nexin-2, and is cleaved by γ-secretase [86], which is also being studied as a potential therapeutic target against breast cancers, including TNBC [87,88,89]. Both proteins are deregulated or overexpressed in different types of cancer [90,91]. Overexpression of APP was found to increase markers of the EMT in breast cancer cells, while silencing of APP had the opposite effect [92]. Many papers in the literature investigate the role of APP in regulating inflammation and angiogenesis in the brain, but very little is known about the role of secreted APP or APLP2 in these processes in the context of cancer, let alone TNBC. One clue may be the cross-reactivity of the RAGE receptor, which binds to several ligands, including both APP and HMGA1 (see the earlier section under “Cytokines and Growth Factors”) [93].

Five different labs reported a constituent of the TNBC secretome called serine peptidase inhibitor E1 (SERPINE1), also known as plasminogen activator inhibitor 1 (PAI1). Ziegler (2016) found SERPINE1 by mass spec analysis of conditioned media from TNBC cells [11]. Dore-Savard (2016) discovered the same protein in the interstitial fluid of xenograft tumors [28]. Other groups found SERPINE1 present in the secretome of TNBC cell lines under specific conditions: when HMGA1 was silenced [25], when the cells were treated with tocotrienols (vitamin E) [60], and when the cells were depleted of LRP-1 [26]. As the alternate name, PAI1, implies, this protein regulates the plasminogen–plasmin system, connecting it to the blood-clotting proteins already discussed.

Small-molecule inhibitors of SERPINE1/PAI1 exist. Two such inhibitors induced apoptosis and disrupted tumor vasculature in xenograft tumors [94]. One of these same small molecules also reduced amyloid-β plaques in the hippocampus and cerebral cortex in a mouse model of Alzheimer’s disease, which resulted in enhanced memory and learning [95]. It does not appear that either of these small molecules has been tested in breast cancer. SERPINE1 plays multiple roles in promoting cancer, including inflammation, which provide both opportunities and challenges in targeting it therapeutically [96].

Treatment of TNBC cells cocultured with endothelial cells with TGFβ was found to stimulate release of the cytokine CCL5 from the endothelial cells and SERPINE1 from the TNBC cells. This created a positive feedback loop that enhanced the metastatic abilities of the tumor cells and that depended on the chemokine receptor CCR5 [97]. Importantly, we note that all three of these proteins (SERPINE1/PAI1, CCL5, and TGFβ) were found in the TNBC secretome.

There are four tissue inhibitor of metalloproteinases (TIMP) family members (TIMP1-4) with similar but overlapping binding partners. They broadly inhibit matrix metalloproteinases (MMPs) by reversible competitive inhibition. TIMPs have also been discovered to play a role in lung branching morphogenesis, adipogenesis, and development of various epithelial and connective tissues [98].

TIMP1 and TIMP2 were discovered in Ziegler et al.’s mass spec screen of TNBC conditioned medium [11]. In addition, TIMP1 was found in the interstitial fluid of xenograft tumors [28], and TIMP1, -2, and -3 were found in medium from TNBC cells depleted of LRP-1 [26].

TIMP1 (also called collagenase inhibitor- and erythroid-potentiating activity) is a glycoprotein with a variety of functions besides inhibiting MMPs [99,100,101]. TIMP1 stimulates the growth of TNBC cells via Akt and NFκB signaling pathways [102]. TIMP1 was also found to induce the EMT via TWIST1 in MCF-10A breast epithelial cells by binding to the Tetraspanin/CD63 receptor. Furthermore, interaction of TIMP1 with Tetraspanin/CD63 occurred even in mutant proteins lacking the MMP inhibitory domain [103]. TIMP1 was also recently found to bind functionally to the invariant chain of MHC class II (CD74), which is involved in various inflammatory diseases [104].

TIMP1 seems to play an important role in breast cancer. Cheng et al. (2016) performed an analysis using the ONCOMINE microarray database and found that TIMP1 was more highly expressed in invasive breast cancer and ductal breast cancer compared to normal breast tissue. The same study found that TIMP1 mRNA and protein is more highly expressed in TNBC cell lines than in non-TNBC cells or normal mammary cells. And TIMP1 levels were elevated in serum from TNBC patients and correlated with poor prognosis [102].

TIMP1 could be a promising target for preventing metastasis; TIMP1 that was incorporated into extracellular vesicles was found to promote metastasis of colorectal cancer to the liver [105], and TIMP1 gene expression was strongly associated with breast cancer metastasis to lymph nodes [106].

The related protein TIMP2 is also important in breast cancer [107]. One study found that suppression of TIMP2 by the protein EZH2 enhanced the invasiveness of TNBC cells by increasing the activity of MMP-2 and MMP-9 [108]. Another study found that single-nucleotide polymorphisms (SNPs) within the TIMP2 gene correlated significantly with breast cancer risk in Han Chinese women [109]. When exogenous TIMP2 was administered to a mouse model of TNBC, tumor growth and metastasis to the lung was reduced, and markers of the EMT, angiogenesis, and pro-metastatic cell signaling were disrupted [110].

TIMP3 was also found in the secretome of TNBC. TIMP3 is downregulated in most cancers and plays a tumor suppressor role in many cancer processes. It is also highly regulated by microRNAs [111]. Intriguingly, TIMP3 KO mice expressing the PyMT or Neu oncogenes under the control of the MMTV promoter exhibited delayed tumor onset or a complete lack of tumor development, respectively [112]. This paradoxical finding is in line with the fact that TIMP3 interacts with the ECM protein EFEMP1/FBLN3 [113] discussed above, which also has pro- and anti-tumor properties.

### 2.4. Other Proteins

#### 2.4.1. Membrane and Extracellular Vesicle Proteins

Extracellular vesicles (EVs) (Table 4) consist of three types of particles of different sizes, including exosomes (30–150 nm diameter), microvesicles (500–2000 nm), and apoptotic bodies (50–5000 nm) [114]. EVs may contain proteins, nucleic acids, and lipids [115].

Ziegler et al. (2016) found that putative exosomal proteins EEF1A1, ENO1, GAPDH, HSPA8, LDHA, MSN, and SDCBP were more highly expressed in TNBC CM on average compared to in MCF-7 CM. Other typical exosomal proteins found in the TNBC secretome included the tetraspanins TSPAN11 and CD151, which are characterized by their four transmembrane domains [117,118]. ITGβ4 was found in exosomes derived from TNBC cells [120] and in a separate study in media from TNBC cells cocultured with CAFs [121]. Importantly, ITGβ4 helps direct the organotropic behavior of metastatic cells [121].

Dong et al. (2022) reviewed the role of EVs in various aspects of TNBC pathobiology, diagnosis, and treatment. EV-associated proteins are involved in TNBC growth; regulation of lymphocytes, macrophages, and fibroblasts in the microenvironment; drug resistance; and metastasis. Specifically, cofilin-1, ITGβ4, ASPH, UCHL1, SPANXB1, and TGFβ1 were found to be involved in TNBC metastasis [5].

#### 2.4.2. Peptide Hormones

Two hormones were found in the TNBC secretome: Follistatin (FST) and Proenkephalin (PENK) (Table 4). FST was first described as an ovarian hormone that binds activin and regulates FSH release from the anterior pituitary. FST is now known to be secreted by many tissues of the body. It also binds to some members of the TGFβ superfamily and modulates the inflammatory response [124]. FST expression is lower in breast cancer compared to normal breast tissue [125], and low expression predicts poor prognosis in TNBC [126].

PENK is the precursor protein for at least two opioid peptide hormones: Leu-enkephalin and Met-enkephalin (opioid growth factor, OGF). OGF treatment inhibited growth of TNBC cells in culture, while protecting them from paclitaxel-induced cell death [127]. PENK itself has been proposed as part of a screening panel for early detection of breast cancer [128].

#### 2.4.3. Metabolic Proteins

It has long been observed that tumor cells are more metabolically dependent on anaerobic glycolysis than normal cells. It is not a surprise, then, that glycolytic enzymes are overexpressed in TNBC cells, and tumors as well (Table 4). Ziegler et al. (2016) found that 11 glycolytic proteins were expressed at higher levels in CM from at least 1 TNBC cell line compared to normal cells. Of these, ENO1, GAPDH, LDHA, and LDHB were more highly expressed in TNBC CM on average compared to in MCF-7 CM [11]. Glycolytic and other metabolic enzymes are common in EVs, which mediate angiogenesis, immune evasion, drug resistance, and activation of macrophages and fibroblasts, leading to invasion and metastasis [129].

In addition to the direct mediators of glycolysis, at least one regulator of glucose metabolism was found in the TNBC secretome. TXNIP belongs to the α-arrestin protein family and interacts with glucose transporters and regulates cellular uptake of glucose [130], although it regulates a variety of other processes as well. A study utilizing TNBC cells cocultured with hBMECS suggested that TXNIP may help brain-seeking TNBC cells disrupt the blood–brain barrier via its blood-vessel-modulating activity [7].

Other metabolic regulators may be important in the TNBC secretome. Lipoprotein A (LPA) was enriched in the conditioned medium of TNBC cells compared to in MCF-7 cells [11]. Normally expressed only in the liver, LPA is closely related to low-density lipoprotein (LDL) and bears some structural similarity to plasminogen (PLG) [131]. In fact, LPA inhibits the activation of plasminogen [132]. LPA is a target of proteolytic cleavage, regulates the plasminogen–plasmin system, and is a regulator of angiogenesis [133,134], reflecting several themes we have already observed in the TNBC secretome.

### 2.5. Drug-Induced Changes

Due to the lack of expression of the estrogen receptor (ER), the progesterone receptor (PR), and amplification of the HER2 protein, chemotherapy has remained the mainstay treatment for TNBC. Chemotherapy is known to alter the secretome, which may in turn promote tumor relapse or resistance to chemotherapy ([19] and reviewed in [4]). While chemotherapy can be effective at killing tumor cells, some cells are resistant to chemotherapy, and chemotherapy can induce “therapeutic induced senescent” (TIS) cells. Interestingly, TIS cells have been found to secrete increased numbers of EVs, which contain activators of proliferation, integrins, and Rab proteins, amongst others, for a total of 142 proteins [135]. While senescent cells are unable to proliferate, they remain metabolically active, allowing for the secretion of proteins and extracellular vesicles, which suggests that senescent cells can impact surrounding cells. The secretions from senescent cells have been studied and collectively termed the senescence-associated secretory phenotype (SASP). There is contradictory information about whether the SASP, which is the secretion of inflammatory cytokines, immune modulators, growth factors, extracellular vesicles, and proteases, induces a pro- or anti-tumorigenic environment. For example, the SASP has been shown to induce cell proliferation and migration of tumor cells, as well as rendering them resistant to chemotherapy. Alternatively, SASP factors may also be responsible for the recruitment of tumor-clearing immune cells, lending to the anti-tumorigenic functions of the SASP [4,136]. In addition to chemotherapy, Palbociclib, which is a targeted therapy used to inhibit CDK4/6, also induces senescence and changes the secretome in TNBC cells [117]. Secreted factors include proteins involved in tumor cell proliferation, drug response, and stemness.

Paclitaxel is another chemotherapeutic agent used in the treatment of TNBC. It is a member of the drug family microtubule-targeting agents (MTAs), which are further subclassified into microtubule-stabilizing and microtubule-destabilizing agents. Treatment with paclitaxel has been shown to induce secretion of IL-8 and enhance the expansion of the cancer stem cell population [137]. Inositol-requiring enzyme 1 alpha (IRE1), which has RNase and kinase activities, has been associated with tumor progression and recently was found to regulate the production of cytokines such as IL-6 and IL-8 after treatment with paclitaxel [19]. The use of a small-molecule inhibitor, MKC8866, that targets the RNase activity of IRE1 results in a decreased production of paclitaxel-induced pro-tumorigenic cytokines. Furthermore, the combination of MKC8866 and paclitaxel resulted in an enhanced response to paclitaxel. These studies suggest that targeting IRE1 may be a therapeutic intervention suitable for reducing the pro-tumorigenic secretome induced by TNBC cells and is further exacerbated by treatment with the chemotherapeutic agent paclitaxel.

Other studies using eribulin, another MTA that is approved for metastatic breast cancer, have shown a post-treatment increase in the secretion of stress-associated proteins, such as growth differentiation factor 15 (GDF15), originally called macrophage inhibitory cytokine-1 (MIC-1) [138]. Increased secretion of GDF15 caused drug resistance in the cells, and blocking GDF15 resensitized the cells to eribulin. In addition to GDF15 induction by eribulin, secreted proteins fell into three STRING pathways post-treatment. One of these networks contained proteins related to cytoskeleton–vesicle trafficking, another contained translation machinery protein, and the third contained proteins linked to cellular stress. All three of these could offer insight into potential biomarkers and therapeutic targets for drug resistance in TNBC. Not surprisingly, these studies further showed that drugs with a similar mechanism of action (eribulin and vinorelbine) induced secretions of similar proteins, as opposed to drugs with different mechanisms of action (i.e., paclitaxel).

## 3. Secreted Factors Outside of the Primary Tumor

While the above section catalogs the secreted factors from TNBC cells, this section will discuss the influence of proteins secreted from the tumor cells on the surrounding cells or tumor microenvironment (TME) and the influence of stromal secretions on the primary tumors. While it is well established that crosstalk between the tumor cells and the TME occurs and can influence the tumorigenic properties, newer information demonstrates that secreted proteins are at least partially responsible for this interaction.

### 3.1. Influence of TNBC on Surrounding Cells

The secretome of TNBC cells may influence the gene expression and secretome of many surrounding tissues. This could be advantageous for the tumors, as it may impact the ability of the tumors to grow and metastasize, leading to the aggressive nature of TNBC. Previous studies have demonstrated changes in the secretome of the ER+ breast cancer cell line, MCF-7, compared to the TNBC cell line, MDA-MB-231. In addition to the subtype specificity, secreted factors of the MDA-MB-231 cells were able to activate mesenchymal stromal cells (MSCs) and further induce activation of macrophages [139]. These data suggest a more aggressive, metastatic, and vascularized tumor environment surrounding the TNBC cells compared to ER+ cells. Interestingly, additional separation was designated using sublines of cells isolated from organ-specific metastases in the MDA-MB-231 model (bone, brain, and lung). The secretome from the lung-, bone-, and brain-specific MDA-MB-231 cells showed organ-specific changes in protein expression, which suggests that a niche may be established for organ-specific metastatic spread [139].

In a study by Hamester et al., the secretomes of three breast cancer cell lines were compared to determine how tumor cells influence the blood–brain barrier (BBB). Using the ER+ cell line (MCF7), the TNBC cell line (MDA-MB-231), and the brain-metastatic-specific derivative of the TNBC line (MDA-MB-231BR) as mentioned above, the authors were able to show that secreted factors from TNBC cells have a greater ability to alter the BBB and aid in breast cancer brain metastatic development [7]. Changes in IL-10 signaling and expression of chemokine receptors were observed after exposure to secreted factors from TNBC cells compared to the ER+ MCF-7 cells, suggesting that this is a specific effect of the TNBC subtype of breast cancer. In addition, secretion of pro-inflammatory molecules from the TNBC cells can prime the endothelium for tumor cell attachment. The impact of the endothelium is also seen in an increase in chemokine secretion from the endothelial cells, which may be working in a paracrine fashion on the tumor cells or in an autocrine fashion on the endothelial cells. Importantly, a difference also exists between the brain-specific MDA-MB-231BR cells and the MDA-MB-231 cells, with the secreted factors from the BR-specific cells inducing IL-6 and IL-7 signaling, as well as increased gap junction assembly [7].

In another example of tumor cell–stroma interaction, Jang et al., demonstrated a positive feedback loop between CD44 and IL-1β. CD44 is secreted specifically from TNBC cells, as MDA-MB-231 and MDA-MB-468 cells but not the ER+ MCF-7 cells, showed secretion of CD44 [22]. Secretion of CD44 from TNBC induces activation of the inflammasome and secretion of IL-1β from surrounding macrophages. To complete the feedback pathway, the IL-1β from the macrophages continues to induce CD44 production from the TNBC cells. Upon blocking CD44 from the TNBC cells, tumor growth is inhibited, suggesting that this feedback mechanism could provide a therapeutic target in the future [22].

### 3.2. Proteins Secreted from the TME

The TME comprises many cell types, including adipocytes, endothelial cells, pericytes, immune cells, fibroblasts, MSCs, osteoblasts, chondrocytes, and components of the extracellular matrix. Some groups will segregate the stromal compartments into various groups (fibroblasts, macrophages, lymphatic endothelial cells, and blood microvascular endothelial cells), while others use broader classifications [24]. Regardless, these cell types have all been shown to influence tumor cells, from primary tumor cell proliferation to metastatic colonization in another organ. It has been long appreciated that secretion of TGF-beta from fibroblasts can activate or inhibit tumor growth, and that cancer-associated fibroblasts (CAFs) secrete specific factors that are distinct from normal fibroblasts (reviewed in [140]). These cells are also able to regulate the accessibility of or responsiveness to chemotherapeutic agents based on the expression of proteins that may be involved in drug metabolism.

Much effort has been devoted to identifying the secreted factors from adipose cells that activate the epithelial-to-mesenchymal transition (EMT) in TNBC cells. In particular, the secretome from mature adipocytes can induce invasive potential and phosphorylation of the transcription factor, STAT3, in MDA-MB-231 cells. The increased phosphorylation of STAT3 provides a potential therapeutic target ([141] and reviewed in [142]), as multiple STAT3 inhibitors have been used in clinical trials for cancer treatment. While some of this interaction is hormone-related (i.e., increased production of leptin and estrogen), adipocytes are also able to secrete cytokines such as IL-6 and IL-8, which are known to increase both tumor initiation and metastatic spread [143,144]. IL-6 can also be produced by the tumor cells and secreted to interact with the IL-6 receptor on lymphatic endothelial cells (LECs) inducing production of the chemokine CCL-5. Upon secretion of CCL-5 from the LECs, tumor cells expressing the receptor for CCL-5 (CCR-5) will be recruited to the LECs and presumably integrated into the lymphatic system for further metastatic colonization [34]. Inhibition of either the CCR-5 or IL-6 receptor resulted in decreased tumor growth and metastatic spread [34], suggesting that there are three points of potential therapeutic intervention in this pathway (STAT3, IL-6, or CCR-5) for TNBC.

A study performed cytokine array analysis of four different TNBC cell lines treated with CM from four different stromal cell populations (lymphatic endothelial cells (LECs), microvascular endothelial cells (MECs), fibroblasts, and macrophages) [24]. In addition to the cytokines involved in the EMT, cell proliferation, and metabolism, a major factor identified as being secreted from the stromal cells was lipocalin 2 (LCN2), which is involved in tumor progression and metastasis. Further studies demonstrated that blocking LCN2 secretion from the stromal cells resulted in decreased TNBC cell proliferation and in vitro metastasis potential [24], and LCN2 inhibitors have been investigated recently as potential targeted therapies [145].

Finally, expression of PIK3C-delta by fibroblasts has also been shown to enhance TNBC progression [12], and inhibition of PIK3C-delta decreases TNBC cell invasion. In addition, expression of PIK3C-delta altered the fibroblast secretome, including PLGF and BDNF, which regulate tumor cell expression of the transcription factor NR4A1, and advanced the aggressiveness of the TNBC population [12].

## 4. Conclusions

By reviewing the current literature on the TNBC secretome, we encountered nearly 80 proteins preferentially expressed by TNBC tumors or cell lines. These molecules encompass a broad spectrum of protein functions, including cytokines, growth factors, ECM proteins, secreted proteases (and their regulators), membrane proteins (either shed from the cell surface or incorporated into extracellular vesicles), peptide hormones, and proteins typically involved in cellular metabolism (Figure 1). In addition, the proteins we encountered are known to be involved in wide-ranging cancer-promoting processes, including tumor growth, angiogenesis, inflammation, the EMT, drug resistance, invasion, and development of the premetastatic niche (Figure 2).

Some of the proteins listed in the tables in this review have already been targeted for therapy, and some even have approved treatments. Our goal was to bring together in one paper all the known targetable proteins within the TNBC secretome and point out the opportunities available for novel therapies against TNBC.

Several themes stand out from this review of TNBC secretome proteins. One strong theme is blood vessel regulation, and especially angiogenesis. The crucial angiogenic protein VEGF seems particularly important in TNBC, although the TGF superfamily seems important, too, though its role with angiogenesis is less clear. Anti-VEGF antibody and small-molecule drugs are already available and will most certainly need to be part of the treatment strategy for TNBC [146], although caution should be taken regarding targeting angiogenesis exclusively since at least one study suggested this may cause rebound tumor progression upon cessation of treatment [147].

Another common theme is immune modulation. A bevy of chemokines, pro-inflammatory proteins, and other cytokines that modulate the immune response were found in the TNBC secretome. These molecules may alter the behavior of endothelial cells to permit metastasis through intra- and extravasation of tumor cells. This was especially evident in the cytokines TXNIP, CXCL1/GRO, IL-1, and others downstream of syndecan-1 that are responsible for TNBC cells crossing the blood–brain barrier (BBB) [7,23,35]. Our review encountered many cytokines, some of which have overlapping functions. Therefore, therapeutic strategies involving inhibition of one or two cytokines may prove insufficient. We recommend a strategy that combines inhibitors or monoclonal antibody treatments against several targets at once. For example, combining immune checkpoint inhibitors with other standard treatments, including antiangiogenic treatments, has been suggested as a winning strategy against TNBC [148,149].

ECM proteins in the TNBC secretome are quite diverse. In addition to their structural and adhesive properties, many ECM proteins have been proven to have unexpected roles in cell signaling and immune system regulation. Some affect processes already discussed above, and some, such as fibulin 1, may alter the response of TNBC to chemotherapy [61]). We are constantly learning about novel functions of ECM proteins and the potential ways they may affect cancer progression.

Many enzymes are present in the TNBC secretome. These proteases are responsible for remodeling the ECM and for activating multiple ECM components. One standout theme in our review is the plasminogen–plasmin system. A number of identified TNBC secreted proteins play some role in activating or regulating this signaling system, which touches on both the inflammatory response and blood vessel dynamics.

Along with the proteases in the TNBC secretome come inhibitors of these enzymes. TIMPs, and other proteins that modify MMPs are well represented among the secreted factors. These proteins likely play a role in regulating tumor invasion.

EVs may be important in TNBC progression as well. EVs are known to be filled with signaling proteins as well as nucleic acids that may alter the behavior of other tumor cells and stromal cells such as fibroblasts, macrophages, and adipocytes [150]. Metabolic regulators and enzymes have an enigmatic role in EVs, though some have been shown to be important in signaling. The EVs in TNBC are reviewed in more detail by Dong et al. [5].

The TNBC secretome can be modified by treatment with drugs including eribulin, paclitaxel, palbociclib, and probably others. Drug treatment may induce a senescence phenotype with an associated change in protein secretions. This phenotype is capable of stimulating tumor cell proliferation, migration, and resistance to chemotherapy.

TNBC cells have an influence on surrounding cells as well, including a role in the activation of mesenchymal stromal cells and macrophages. These changes lead to changes in the tumor microenvironment (TME) that encourage invasion and metastasis. Furthermore, they may even affect the ability of TNBC cells to extravasate into various metastatic sites, including crossing the BBB.

Proteins secreted from the TME outside of the TNBC cells themselves also play a role in tumor progression. Secreted factors from adipocytes, fibroblasts, macrophages, and endothelial cells have been investigated. In addition to cytokines IL-6, IL-8, and CCL5, LCN2 was identified as a common cytokine factor secreted by stromal cells [24].

Some drugs and biologicals have already been created to target specific proteins secreted by TNBC. Future treatments may require a multi-pronged approach that blocks several secreted proteins or cancer processes at once. It is our hope that this review will spur the development of innovative approaches to treating this aggressive disease that take advantage of the various protein factors found in the TNBC secretome.

## Figures and Tables

**Figure 1 ijms-24-02100-f001:**
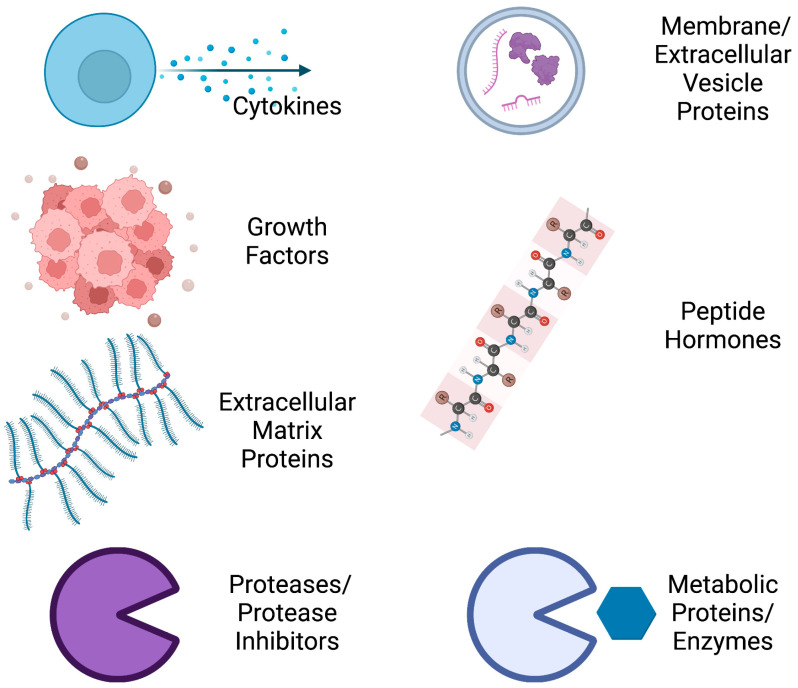
Proteins of the TNBC secretome. A large variety of functional proteins are secreted from TNBC cells and tumors (Created with BioRender.com).

**Figure 2 ijms-24-02100-f002:**
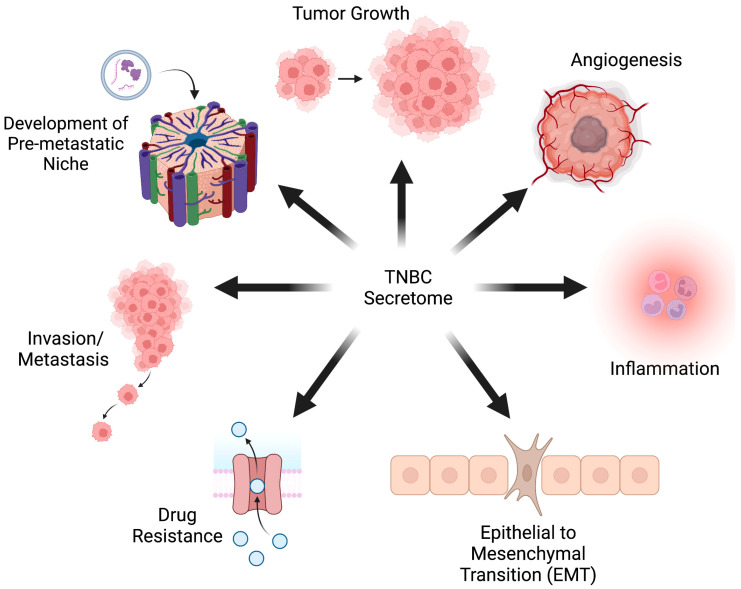
Effects of the TNBC secretome. Protein factors secreted by TNBC cells and tumors regulate cancer-promoting processes (Created with BioRender.com).

**Table 1 ijms-24-02100-t001:** Cytokines and Growth Factors of the TNBC Secretome.

Cytokines				
Protein	Aliases	Source	Evidence	Refs.
CCL2	Chemokine (C-C motif) ligand 2	Adipose-derived MSCs	ADMSCs treated with TNBC CM	[6]
CCL5	Chemokine (C-C motif) ligand 5, RANTES	Adipose-derived MSCs	ADMSCs treated with TNBC CM	[6]
COX2	Cyclooxygenase 2, Prostaglandin synthase 2	Adipose-derived MSCs	ADMSCs treated with TNBC CM	[6]
CXCL1/GRO	CXC motif chemokine ligand 1, growth-related oncogene	TNBC cells	Secretome profiling of CM; TNBC cells cocultured with hBMECS; inhibition of IRE1 in TNBC cells	[7,18,19]
GM-CSF	Granulocyte-macrophage colony-stimulating factor	TNBC cells	Inhibition of IRE1 in TNBC cells	[19]
HIF-1α	Hypoxia-inducible factor-1 subunit alpha	Adipose-derived MSCs	ADMSCs treated with TNBC CM	[6]
HMGA1	High-mobility group A1	TNBC cells and tumors	TNBC tumors	[20,21]
IL-1β	Interleukin-1 beta	Macrophages, adipose-derived MSCs, TNBC cells	CM and patient serum in mouse model; ADMSCs treated with TNBC CM; TNBC cells in model of blood–brain barrier	[6,22,23]
IL-6	Interleukin-6	TNBC cells; adipose-derived MSCs	Secretome profiling of CM; ADMSCs treated with TNBC CM; Inhibition of IRE1 in TNBC cells	[6,18,19]
IL-8	Interleukin-8	TNBC cells	Secretome profiling of CM; Inhibition of IRE1 in TNBC cells	[18,19]
LCN2	Lipocalin-2	Stromal cells	TNBC cells cocultured with four types of stromal cells	[24]
PD-L1	Programmed death ligand 1	Adipose-derived MSCs	ADMSCs treated with TNBC CM	[6]
PLAUR	suPAR, soluble urokinase receptor	TNBC cells	TNBC cells (inducible silencing of HMGA1)	[25]
SAA1	Serum amyloid A1	TNBC cells	MS analysis of CM	[11]
SLIT3	Slit guidance ligand 3	Fibroblasts	TNBC cells treated with CAF CM	[15]
THBS1	Thrombospondin 1	TNBC cells	MS analysis of CM	[11]
**Growth Factors**				
BDNF	Brain-derived neurotrophic factor	Fibroblasts	Coculture of fibroblasts with TNBC cells	[12,13]
END1	Endothelin-1	TNBC cells	CM from TNBC cell lines depleted of Syndecan-1	[16]
GRN	Granulin	TNBC cells	TNBC cell line depleted of LRP-1	[26]
NODAL	Nodal growth differentiation factor (TGFβ superfamily)	TNBC tumors	IHC of human tumors	[27]
PLGF	Placental growth factor	Fibroblasts	Coculture of fibroblasts with TNBC cells	[12]
TGFβ1	Transforming growth factor beta 1	TNBC cells	TNBC cell line depleted of LRP-1; MS analysis of CM	[11,26]
TGFβ2	Transforming growth factor beta 2	TNBC cells	Inhibition of IRE1 in TNBC cells	[19]
VEGF-A	Vascular endothelial growth factor A	TNBC cells and xenograft tumors; Adipose-derived MSCs	CM from TNBC cell lines depleted of Syndecan-1; interstitial fluid of xenograft tumors; ADMSCs treated with TNBC CM	[6,16,28]

**Table 2 ijms-24-02100-t002:** Extracellular Matrix Proteins of the TNBC Secretome.

Protein	Aliases	Source	Evidence	Refs.
BGN	Biglycan	TNBC cells	MS analysis of CM	[11]
CD44	Cluster of differentiation 44	TNBC cells	MS analysis of CM; CM and patient serum in mouse model	[11,22]
CD109	Cluster of differentiation 109	TNBC cells	MS analysis of CM	[11]
DAG1	Dystroglycan	TNBC cells	MS analysis of CM	[11]
DCN	Decorin	TNBC cells	MS analysis of CM	[11]
ECM1	Extracellular matrix protein 1	TNBC cells	MS analysis of CM; TNBC cell line depleted of LRP-1	[11,26]
EFEMP1/FBLN3	EGF-containing fibulin-like extracellular matrix protein 1, Fibulin 3	TNBC cells	MS analysis of CM	[11]
FBLN1	Fibulin 1	CAFs	TNBC cells treated with CAF CM	[15]
FMOD	Fibromodulin	TNBC cells	MS analysis of CM	[11]
IGFBP4	Insulin-like growth factor-binding protein 4	TNBC cells	MS analysis of CM	[11]
IGFBP7	Insulin-like growth factor-binding protein 7	TNBC cells	MS analysis of CM	[11]
IGFBP10/ Cyr61/CCN1	Insulin-like growth factor-binding protein 10, Cysteine-rich angiogenic inducer 61, CCN1	TNBC cells (exosomes)	Migration of TNBC cells decreased by neutralizing antibodies	[36]
L1CAM	L1 cell adhesion molecule	TNBC cells	MS analysis of CM	[11]
LGALS1	Galectin 1	TNBC cells	MS analysis of CM	[11]
LGALS3BP	Galectin 3-binding protein	TNBC cells	MS analysis of CM	[11]
LOXL2	Lysyl oxidase-like 2	TNBC cells	MS analysis of CM	[11]
LTBP1	Latent TGFβ-binding protein 1	TNBC cells	MS analysis of CM	[11]
NRCAM	Neuronal cell adhesion molecule	TNBC cells	MS analysis of CM	[11]
P4HB	Protein disulfide-isomerase, prolyl 4-hydroxylase beta	TNBC cells	MS analysis of CM	[11]
PLOD1	Procollagen-lysine, 2-oxoglutarate 5-dioxygenase 1	TNBC cells	MS analysis of CM	[11]
PPIB	Peptidyl-prolyl cis-trans isomerase B	TNBC cells	MS analysis of CM	[11]
RELN	Reelin	TNBC cells	TNBC cells (knockdown of integrin alpha3)	[37]
TF	Tissue factor	TNBC cells	CM from TNBC cell lines depleted of Syndecan-1	[16]
TLN1	Talin 1	TNBC cells	MS analysis of CM	[11]
TNC	Tenascin C	TNBC cells	MS analysis of CM	[11]

**Table 3 ijms-24-02100-t003:** Proteases and Protease Inhibitors of the TNBC Secretome.

Protein	Aliases	Source	Evidence	Refs.
CTSD	Cathepsin D	TNBC cells	TNBC cells treated with tocotrienols (vitamin E)	[60]
CTSZ	Cathepsin Z	TNBC cells	MS analysis of CM	[11]
GGH	Gamma-glutamyl hydrolase	TNBC cells	MS analysis of CM	[11]
PCSK9	Proprotein convertase subtilisin/kexin type 9	TNBC cells	MS analysis of CM	[11]
PLAT	Tissue plasminogen activator	TNBC cells	TNBC cell line depleted of LRP-1	[26]
PLAU	uPA, urokinase plasminogen activators	TNBC cells; xenograft tumors	TNBC cells (inducible silencing of HMGA1); interstitial fluid of xenograft tumors	[25,28]
PLG	Plasminogen	TNBC cells	TNBC cell line depleted of LRP-1	[26]
APLP2	Amyloid-beta precursor-like protein 2	TNBC cells	MS analysis of CM	[11]
APP	Amyloid-beta precursor protein	TNBC cells	MS analysis of CM	[11]
PI3	Elafin, peptidase inhibitor 3	TNBC cells	MS analysis of CM	[11]
SERPINE1	Serine protease inhibitor E1, plasminogen activator inhibitor 1 (PAI1)	TNBC cells; xenograft tumors	MS analysis of CM; interstitial fluid of xenograft tumors; TNBC cells (inducible silencing of HMGA1); TNBC cells treated with tocotrienols (vitamin E); TNBC cell line depleted of LRP-1	[11,25,26,28,60]
TIMP1	Tissue inhibitor of metalloproteinases 1	TNBC cells; xenograft tumors	MS analysis of CM; interstitial fluid of xenograft tumors	[11,28]
TIMP2	Tissue inhibitor of metalloproteinases 2	TNBC cells	MS analysis of CM	[11]
TIMP1, -2, -3	Tissue inhibitor of metalloproteinases	TNBC cells	TNBC cell line depleted of LRP-1	[26]

**Table 4 ijms-24-02100-t004:** Other Proteins of the TNBC Secretome.

Membrane and Extracellular Vesicle Proteins				
Protein	Aliases	Source	Evidence	Refs.
ANXA2	Annexin II, annexin A2	TNBC cells	TNBC cell lines	[116]
BCAP31	B cell receptor-associated protein 31	TNBC cells	TNBC cell line treated with Palbociclib	[117]
CD151	Cluster of differentiation 151	TNBC tumors	Exosomes from TNBC patient serum	[118]
IL-3Rα	Interleukin-3 receptor subunit α	TNBC tumors and cells	TNBC tumor expression and modulation of IL-3R-containing EVs	[119]
ITGB4	Integrin β4	TNBC cells	TNBC cells in coculture with CAFs; exosomes from TNBC cells	[120,121]
SPANXB1	Sperm protein associated with the nucleus on the X chromosome B1	TNBC tumors	Circulating EVs from TNBC patients	[122]
TSPAN11	Tetraspanin 11	TNBC cells	TNBC cell line treated with Palbociclib	[117]
UCHL1	Ubiquitin carboxyl-terminal hydrolase isozyme L1	TNBC tumors and cells	Exosomes, serum, and CM	[123]
**Peptide Hormones**				
FST	Follistatin	TNBC cells	TNBC cell line depleted of LRP-1	[26]
PENK	Proenkephalin	CAFs	TNBC cells treated with CAF CM	[15]
**Metabolic Proteins**				
ENO1	Alpha-enolase	TNBC cells	MS analysis of CM	[11]
GAPDH	Glyceraldehyde-3-phosphate dehydrogenase	TNBC cells	MS analysis of CM	[11]
LDHA	Lactate dehydrogenase A	TNBC cells	MS analysis of CM	[11]
LDHB	Lactate dehydrogenase B	TNBC cells	MS analysis of CM	[11]
LPA	Lipoprotein A	TNBC cells	MS analysis of CM	[11]
TXNIP	Thioredoxin-interacting protein	TNBC cells	TNBC cells cocultured with hBMECS	[7]

## Data Availability

No new data were created or analyzed in this study. Data sharing is not applicable to this article.

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
