# Peer review of "Proteins Found in the Triple-Negative Breast Cancer Secretome and Their Therapeutic Potential"

_ijms, 2023, doi:10.3390/ijms24032100_

Round 1

Reviewer 1 Report

This manuscript describes the various secretomes of triple negative breast cancer and their therapeutic potential. The manuscript is well organized as a review article, and is considered to be an important data for presenting various proteins secreted from TNBC, their roles, and future treatment directions. Therefore, this paper is considered suitable for publication in International Journal of Molecular Sciences.

Author Response

Thank you. We tried to present the most important facts and organize them clearly.

Reviewer 2 Report

The authors have written a well summarized review on proteins secreted by TNBC cancers and their therapeutic applications and options. TNBC cancer indeed is a very challenging subtype of breast cancer and has limited treatment options. I appreciate the authors for their excellent review.

However, I would like the authors to suggest highlighting the CD44 receptor and carcinoembryonic antigen receptor expressed by the TNBC cells which could serve as an excellent targeted therapeutic delivery route. Recent articles such as https://doi.org/10.1021/acs.molpharmaceut.2c00439; https://doi.org/10.1021/acsami.1c21655 could be referred for this context.

Author Response

We did not pay enough attention to CD44. Several sentences have been added (starting at line 162) that elaborate more on the role of CD44 in breast cancer, its connection with CEA, and the new nanoparticle-based approaches that target these proteins. Thank you very much for the tip!